# Stop DDoS Attacking the Research Community with AI-Generated Survey Papers

**Jianghao Lin**[1], **Rong Shan**[2], **Jiachen Zhu**[2], **Yunjia Xi**[2], **Yong Yu**[2], **Weinan Zhang**[2]*
[1]Antai College of Economics and Management, Shanghai Jiao Tong University, China
[2]School of Computer Science, Shanghai Jiao Tong University, China
{linjianghao,wnzhang}@sjtu.edu.cn

## Abstract

Survey papers are foundational to the scholarly progress of research communities, offering structured overviews that guide both novices and experts across disciplines. However, the recent surge of AI-generated surveys, especially enabled by large language models (LLMs), has transformed this traditionally labor-intensive genre into a low-effort, high-volume output. While such automation lowers entry barriers, it also introduces a critical threat: the phenomenon we term the *"survey paper DDoS attack"* to the research community. This refers to the unchecked proliferation of superficially comprehensive but often redundant, low-quality, or even hallucinated survey manuscripts, which floods preprint platforms, overwhelms researchers, and erodes trust in the scientific record. In this position paper, we argue that **we must stop uploading massive amounts of AI-generated survey papers (*i.e.*, survey paper DDoS attack) to the research community,** by instituting strong norms for AI-assisted review writing. We call for restoring expert oversight and transparency in AI usage and, moreover, developing new infrastructures such as *Dynamic Live Surveys*, community-maintained, version-controlled repositories that blend automated updates with human curation. Through quantitative trend analysis, quality audits, and cultural impact discussion, we show that safeguarding the integrity of surveys is no longer optional but imperative to the research community.

## 1 Introduction

In today's fast-moving research landscape, survey papers serve as vital waypoints for synthesizing vast literatures, distilling key trends, and pointing the way forward for both newcomers and experts. Over the past few years, the number of survey manuscripts on preprint servers such as arXiv has grown exponentially, driven in large part by the advent of powerful generative AI, especially large language models (LLMs), that can auto-draft literature reviews in minutes [25, 28, 29, 24, 17, 12, 27]. What was once a labor-intensive exercise in critical synthesis has now become a low-barrier, high-volume output bolstered by large language models and automated summarization AI tools.

Yet this very convenience has spawned a new threat to the research community, which is referred to as the "**survey paper DDoS attack**" in this paper. Much like a distributed denial-of-service attack [19, 20], an unchecked flood of AI-generated surveys can overwhelm researchers with redundant overviews, surface unverified or even hallucinated citations, and drown out those genuinely insightful contributions. As the volume of superficially polished AI-generated survey manuscripts grows, the true functionality of a survey, *i.e.*, to critically appraise, compare, integrate, and inspire research, is at risk of being reduced to rote aggregation.

The implications for research quality and trust are profound. First, genuine advances risk being obscured by algorithmically generated rehashes of existing work. Newcomers and interdisciplinary

---

*Corresponding author.

39th Conference on Neural Information Processing Systems (NeurIPS 2025) Position Paper Track.

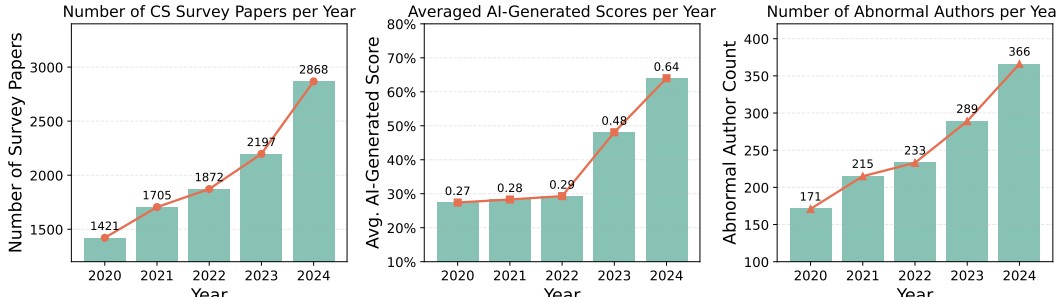

Figure 1: *Left:* The number of CS survey papers over the past 5 years. *Middle:* Averaged AI-generated scores of CS survey papers for the recent 5 years. *Right:* The number of abnormal authors detected per year. The data all witness a post-2022 spike, which is marked by the advent of ChatGPT and other advanced LLMs.

scholars may struggle to locate dependable overviews amid the noise. Moreover, errors or biases introduced by automated drafting can propagate unchecked, seeding subsequent research with faulty premises. In sum, the flood of non-peer-reviewed AI-generated surveys endangers both the rigor of literature reviews and the credibility of the scientific record.

Based on the discussion above, we give our core position: **We must stop uploading massive amounts of AI-generated survey papers (*i.e.*, survey paper DDoS attack) to the research community,** by forcing the correct usage of AI for survey writing, restoring rigorous human oversight, and introducing firm standards for AI-assisted reviews.

The remainder of the paper is organized as follows, which is also illustrated in Figure 2. In Section 2, we present a quantitative analysis of the recent explosion in AI-driven survey submissions on arXiv. Section 3 delves into key quality concerns, including redundancy, citation accuracy, and superficial taxonomy design. Section 4 examines the broader impact of AI-generated survey abuse on research culture and trust. In Section 5, we offer concrete policy suggestions targeting AI-generated survey papers. Section 6 outlines our vision for Dynamic Live Surveys as a sustainable, community-driven alternative to one-off, AI-generated overviews. Finally, we discuss alternative views in Section 7 and conclude in Section 8.

## 2    AI-Generated Survey Paper Explosion on arXiv

To better understand the trend of survey papers, we conduct empirical studies to analyze arXiv submissions from 2020 to 2024 across all computer science (CS) categories, resulting in a total of 10,063 papers. Specifically, we collect submissions whose titles contain keywords such as "*survey*", "*review*", "*overview*", or "*taxonomy*", which empirically indicates the work is a survey paper. Based on the collected survey papers, we perform the following empirical analysis:

- We analyze the trend of the number of survey papers per year, regardless of AI-generated factors.
- We leverage an open-source AI-content detector[2] to estimate the AI-generated scores of these survey papers. The higher the score the survey paper receives, the more likely the paper is generated or assisted by AI.
- We take a look at the number of abnormals who submitted over 3 survey papers in one month with less than 2 collaborators, which could be a strong indicator for AI-generated surveys.

We report the results in Figure 1, from which we obtain the following observations.

**Growth Trends.**    The number of CS survey papers on arXiv has been growing exponentially in recent years, whose volume has become difficult for researchers to keep up with. Figure 1(a) shows that, starting from 2020, the number of survey papers significantly increases year by year, with a noticeably accelerated turning point appearing around 2022–2023. This aligns with the release and widespread adoption of advanced large language models (LLMs), such as OpenAI's ChatGPT [18] in late 2022, and later models like Claude [4] and Google's PaLM [7] and Gemini [21] in 2023. A

---

[2]https://huggingface.co/desklib/ai-text-detector-v1.01

separate arXiv-based study [25], which also focuses on LLM-themed surveys, also confirms that the paper count is increasing rapidly in recent years. It finds that the number of LLM-themed surveys grew steadily from 2019 to 2021 with a sharp rise in 2022 and 2023, which is consistent with our empirical study. Moreover, this upward trend is continuing and possibly accelerating.

**AI-driven Post-2022 Spike.** As shown in Figure 1(b), there is clear evidence of a submission spike after 2022, aligning with the public release of ChatGPT. Some researchers [15] analyze the linguistic patterns in millions of scientific paper abstracts, and conclude that by 2024 over 10% of scientific abstracts were processed by LLMs. A recent large-scale study by an AI content detection company [26] reported a 72% increase in papers that may have been written with the help of AI on arXiv since the availability of ChatGPT. Moreover, the number of papers with high AI-generated content scores roughly doubled, from about 3.6% in late 2022 to around 6.2% by the end of 2023. In our context, a similar trend for AI-driven post-2022 spike is observed when focusing specifically on CS survey papers, as evidenced by the results in Figure 1(b). Taken together, the timeline suggests that the release of ChatGPT and other advanced LLMs plays a major role in boosting AI-generated survey papers. We provide more empirical evidence using other metrics in Appendix A.

**Submission Patterns.** As shown in Figure 1, the number of abnormal authors keeps increasing year by year, also with a slightly accelerated turning point at year 2022. A closer look at arXiv data shows signs that some survey papers may have been automatically generated or largely assisted by AI. In particular, we observe cases where single authors or small teams submit several survey papers in a short period, often covering different topics. This raises doubts about how such papers could be written so quickly. In some of these cases, the authors have little or no previous works in the field they are reviewing. Even more, some survey preprints are even listed under anonymous groups or collectives with no clear link to any institution. These unusual authorship and submission patterns point to the possibility that some AI-generated surveys are being created through organized efforts, perhaps to boost citation counts or fill out CVs, rather than for genuine academic contribution.

As a result, the rapid growth of survey papers is clear across all CS categories on arXiv. In areas like natural language processing (NLP), dozens of "A Survey of X" papers are being published within just a few months, which is far more than what used to be normal [30]. For a more concrete example, we observe over five review papers for "Model Context Protocol (MCP)" [3] in around one month. Although MCP is a surging topic in the past few months, over five released review paper preprints within a very short time are absolutely redundant, potentially confusing the researchers and harming the community.

In summary, the volume of potentially AI-generated surveys has become so overwhelming that even experienced researchers can have difficulty keeping up with the coming surveys. This creates an ironic situation: **using AI to generate surveys without restraint ends up being counterproductive**. Instead of helping the research community to digest a field, it starts to feel like a DDoS attack, flooding the field with more content than anyone can reasonably read or use. We provide more explanation on the link between DDoS attack and survey papers in Appendix B.

## 3 Quality Concerns & Detection Methods

Quantity aside, a core issue is the terrible quality of many AI-generated surveys. Traditional survey papers are valued for synthesis, insight, and authority. They typically:(a) propose a novel taxonomy or framework to organize prior work; (b) provide deep analysis and critique of the state-of-the-art; (c) identify open challenges and future directions; (d) ensure citation diversity and accuracy, giving credit to foundational works and recent advances alike. By contrast, suspected AI-generated surveys often lack most of these elements. In this section, we first point out several problems and concerns about AI-generated surveys. Based on them, we further provide some heuristic metrics to detect potentially AI-generated surveys.

### 3.1 Quality Concerns

**Structural Differences.** SurveyForge [28] shows that AI-generated surveys have explicit structural deficiencies compared with human-written surveys. Specifically, AI-generated surveys tend to have disordered outlines that do not reflect the conceptual structure of the field. They often read like

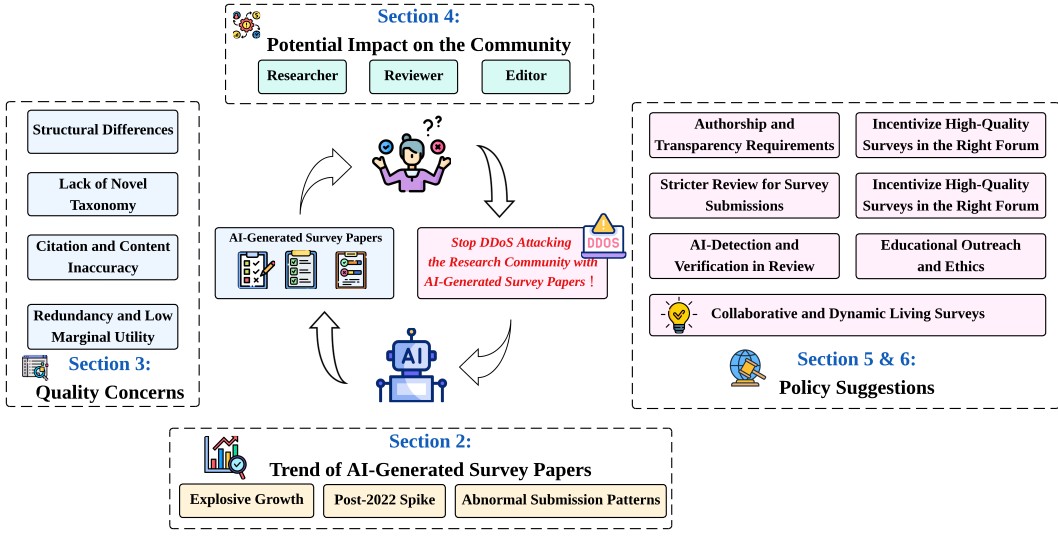

Figure 2: The structure of our position paper.

unorganized enumerations of topics or papers, without a clear narrative flow. Key sections in them (*e.g.*, background, thematic taxonomy) may be shallow or missing. In contrast, a human-written survey on a topic will usually define precise sub-categories and transitions.

**Lack of Novel Taxonomy.** In our empirical study, we find that many suspect surveys simply mimic existing classifications from other sources (even from the Wikipedia entries of the topic), instead of proposing any new way to conceptualize the domain. For instance, multiple AI-written surveys on Vision Transformers (ViT) [8] all split the field into similar sections, *e.g.*, "Backbone Architectures" and "Applications in Classification/Detection". These surveys often resemble each other closely, showing little originality. This suggests a template-based process, where the LLMs possibly rely on the same prominent papers or earlier surveys as guides. In contrast, a well-crafted human-written survey might introduce a new taxonomy, *e.g.*, categorizing ViT by efficiency strategies. The lack of such original structure in many recent survey preprints raises concerns that they may have been generated by AI with limited human insight.

**Citation and Content Inaccuracy.** Perhaps the most notable problem lies in references and factual content. AI-generated surveys frequently show citation anomalies. As noted in [28], these papers often miss truly relevant and influential works, while over-citing less relevant or obscure papers. This suggests that the reference list was assembled by keyword matching rather than expert judgment. In some cases, references appear to be fabricated or incorrect, which typically arises from LLM hallucination [13]. Kobak et al. [15] report that even strong LLMs like ChatGPT sometimes produce fake or incorrect citations, underscoring the risk of using LLMs for scientific papers without proper checks. Indeed, volunteers curating lists of suspect papers (*e.g.*, Academ–AI [2]) have found numerous examples where a preprint's reference list contains works that cannot be found or irrelevant citations that do not align with the context. Such anomalies betray a lack of true scholarly curation, which is seldom seen in surveys written by experienced researchers.

**Redundancy and Low Marginal Utility.** Another concern is the redundancy across these AI-generated surveys. We observe significant overlap in content between different survey papers on the same topic, often with near-identical phrasing. This points to a broader problem of text reuse. When multiple authors ask an LLM to "write a literature review on X," the model often produces very similar responses, especially for common definitions or well-known facts. Recent research [15] has shown a sharp rise in the use of certain writing patterns linked to LLMs, suggesting many papers now share the same style. As a result, these AI-generated surveys often end up repeating the same basic content, without offering new insights. The marginal scholarly value of the N-th survey on a hot topic becomes almost zero, while each still adds to the noise that researchers must filter through.

### 3.2 Detection Metrics

**GPT-generated Phrases.** One simple approach is searching for known GPT-generated phrases in arXiv papers. We write scripts to scan arXiv CS survey submissions for explicit phrases that reveal AI involvement, such as "as an AI language model", "my knowledge cutoff", and "as of September 2021". This does find several matches, clearly showing that the authors had not properly edited the LLM-generated text. In many cases, these obvious signs are the easiest ones to spot.

**Citation Overlap.** Since an LLM might have a certain set of prominent papers it always cites for a topic, we hypothesize that if multiple survey papers of different authors share an unusually high fraction of identical references, it could signal a common AI-generated source. We analyze citation lists from 10 recent surveys on an ML topic and find that, on average, any two share around 60–70% of references. This is higher than one might expect if authors working independently are curating references based on personal literature searches. This suggests a degree of convergence, potentially resulting from shared reliance on similar datasets or summaries. While not definitive proof, a high Jaccard similarity between citation lists may serve as a useful indicator warranting further examination, which is provided in Appendix A.

**Length and Repetition Patterns.** We also investigate the length and repetition patterns of survey papers. AI-generated text can sometimes exhibit over-usage of filler words or repetitive transitions. Using a simple language model, we measure the entropy of word distribution in suspect vs. known-human survey papers. The suspect papers often have lower lexical diversity (*i.e.*, they repeated common phrases more often). Qualitatively, several papers contain successive paragraphs that start identically with "Furthermore" in GPT-written essays. Automated detection of such stylistic signs can be an interesting area of research.

In summary, AI-generated surveys are currently far from satisfactory, lacking elements from low-level structural rigor to high-level critical insight. These issues undermine scholarly standards and highlight the need for the research community to develop more advanced and reliable detection methods.

## 4 Impact on Research Culture and Trust

Beyond immediate quality issues, the flood of AI-generated survey papers has broader impacts on research culture, information credibility, and the scientific direction. In this section, we assess these impacts from the perspectives of researchers, reviewers, and editors, respectively.

**Researcher Perspectives.** Many researchers have voiced increasing frustration with what could be called "**literature clutter**". When conducting reviews for their own work, they often have to sift through a large number of overlapping survey papers. A common complaint is that searching for a topic now yields many nearly identical surveys on arXiv, often of uncertain quality. This is particularly confusing for early-career researchers, who may struggle to tell which surveys are trustworthy or authoritative. A study [23] asked 1,600 researchers about their use of ChatGPT in writing and their views on AI-generated academic work. While many reported experimenting with it, a large number also expressed doubts about the accuracy and completeness of AI-written articles. This skepticism points to a potentially growing concern, where researchers are starting to question whether a given survey, as an important type of research articles, is a serious academic effort or simply an automatically generated summary. If this trend continues, even high-quality survey papers may suffer from reduced trust.

**Reviewer Perspectives.** The surge of AI-generated survey papers has led to **information overload for reviewers**, and increases their **reviewing burden**. A recent study [6] finds that there exists a big volume of AI-written reports that require extra effort to verify. The reviewers reported spending time verifying references in suspicious survey submissions, which could have been spent for evaluating substantive contributions. This inefficiency essentially wastes effort, leading to a drag on the research.

Moreover, the AI-generated survey papers have also caused a **psychological shift**, triggering a negative impact on the reviewing experience of reviewers. Nowadays, on many social media platforms, reviewers express their frustration when they suspect an author hasn't put in effort because the paper is likely mostly AI-generated. It is one thing to review a poorly written paper by a student,

in which case one can give constructive feedback. It is another to review a paper that reads like empty language, which greatly undermines reviewing enthusiasm.

**Editor Perspectives.** On the editorial side, journal editors are increasingly on alert for AI-written content. The guidelines from top-tier journals state that authors must disclose AI assistance and that no LLMs will be accepted as valid co-authors, as they are unable to take responsibility for the work [5, 14]. The same opinion was echoed by Science Magazine's Editor-in-Chief H. Holden Thorp, who famously titled his editorial "ChatGPT is fun, but not an author." [22]. Thorp argued that allowing AI to claim authorship or unchecked contribution reduces scientific transparency and complicates accountability. Furthermore, many conference organizers have put out statements warning against AI-generated submissions. The general editorial view is that while AI can be a useful tool for editing and translation, the core scholarly contributions must be human-originated. Otherwise, we risk diluting the meaning of authorship and publication.

**Impact on Literature Quality and Research Direction.** One of the most serious risks is that it could quietly **shift how research moves forward**. Surveys are meant to guide researchers, showing what has been done and where the gaps are. But if these surveys become unreliable or just repeat the same points, they may give a false picture of the field. Researchers might wrongly assume that a topic is already well studied and not worth revisiting, even if that's not true. Even worse, many AI-written surveys don't clearly point out open challenges or unanswered questions, which might lead students or early-career researchers to believe there's nothing left to explore. In this way, a large number of low-quality surveys can blur the actual state of knowledge and mislead the research community.

Another concern is the potential for **citation distortion**. AI-generated surveys, especially those easily found on platforms like arXiv, may gain citations simply because they are accessible and convenient to reference. Over time, these surveys might begin citing each other, creating a self-reinforcing loop that boosts the visibility of low-quality or less relevant works, while more meaningful and foundational papers get overlooked. This is somewhat similar to the idea of **data poisoning**. In our context, we might call it **literature poisoning**, where noisy or low-value papers distort the citation landscape. As a result, we could see survey papers with many citations despite offering little real insight, or even spreading incorrect information. This not only weakens the quality of academic knowledge but also risks damaging trust. As a previous work [11] pointed out, AI-generated papers that look scientific can slip into academic search engines like Google Scholar. Once their presence is widely known, they may lead readers to question the credibility of all research by default.

In conclusion, the cultural impact of this trend is primarily negative at present. It burdens the community, dilutes the literature, and forces us to question the authenticity of academic work. To preserve the integrity of research, proactive measures are needed, which we turn to next.

## 5 Policy Suggestions for the Research Community

In this section, for the good of broad research community, we propose several policy measures and best practices that could mitigate the "DDoS of AI-generated surveys" while embracing the positive aspects of AI assistance.

**Authorship and Transparency Requirements.** The research community, including major conferences and journals, should require authors to clearly disclose any use of large language models (LLMs) in the writing process. This could be done through a footnote or a short note in the Methods section, specifying what content was generated (*e.g.*, text, code, or figures) and how it was reviewed or revised. While enforcing full honesty may be difficult, having clear guidelines sets shared expectations. In addition, LLMs should not be listed as authors, and authorship should remain limited to accountable humans [22]. However, perhaps a separate "AI Contribution" section could be allowed (*e.g.*, "This paper utilized GPT-4 to draft an initial literature overview, which was then substantially revised by the authors."). This level of transparency would help normalize the responsible use of AI as a tool while discouraging the undisclosed generation of entire papers.

**Stricter Review for Survey Submissions.** For conferences and journals that accept survey papers, the research community should apply a higher review standard to ensure quality. We recommend assigning at least one senior reviewer or area chair to specifically assess the depth and value of

each survey. The review form could include custom questions, *e.g.*, "Does this survey introduce new insights or a meaningful taxonomy?" and "Does it properly cite key foundational works?" Surveys that simply summarize existing material without offering analysis or synthesis should be rejected or redirected to tutorial tracks. In short, the community should adopt a principle similar to a "survey quality threshold", which is not a fixed limit on the number of surveys accepted, but a clear expectation that surveys must provide more value than typical research papers. Poor-quality surveys can mislead a broad audience, making them potentially more harmful than a low-impact research paper.

**Limits on Redundant Submissions.** The community might consider mechanisms to discourage multiple repetitive surveys on the same topic. For example, major conferences and journals could coordinate to avoid turning "survey tracks" into low-bar pathways for easier or faster publication. If one good survey on say, "Foundation Models," was published this year, there is little reason to have five more in the next year unless they bring new angles. While we don't advocate formal bans on topics, program committees can apply stricter judgment when assessing survey papers that don't sufficiently distinguish themselves from existing ones. Additionally, arXiv moderators and possibly conference organizers could flag when an author submits an excessive number of surveys in a short period. Perhaps a gentle warning or inquiry can be sent in such cases to ensure the author affirms the work's integrity.

**AI-Detection and Verification in Review.** The community could invest in AI-generated text detection for screening submissions, and AI-detection could be run as one factor for accessing submissions. A high AI-content score wouldn't mean automatic rejection, since false positives are possible, but it could trigger closer review. Additionally, reviewers can be encouraged to do quick checks, *e.g.*, by verifying a few random references in a survey paper to see if they are accurate and relevant. If obvious signs of AI misuse are found, such as fake citations or phrases like "as of my knowledge cutoff", the paper should be rejected for not meeting academic standards. To make expectations clear, conferences and journals could state in their submission guidelines: "Papers containing AI-generated content that is not properly disclosed may be rejected for violating ethical policies." Clear rules like this could help discourage authors from submitting unedited AI drafts.

**Incentivize High-Quality Surveys in the Right Forum.** One reason people resort to arXiv surveys is the lack of formal venues that welcome survey papers. Many top journals do, but conferences typically do not. The community might consider creating a dedicated and well-curated outlet for surveys, *e.g.*, a "Journal of ML Surveys and Synthesis" or a workshop series. If such a venue had strong peer review and academic recognition, it would encourage authors to invest real effort in producing high-quality work, rather than uploading hastily written, AI-generated surveys to arXiv for possible citation padding. For example, introducing a "Best Survey Paper Award" would help reward surveys that provide meaningful insights. This kind of positive reinforcement can shift the norm, showing that writing a survey is not about hitting a word count with GPT, but about contributing to the community's understanding.

**Educational Outreach and Ethics.** Finally, it should be careful for conferences and universities to educate upcoming researchers on the proper use of LLMs and the pitfalls of AI-generated papers. Many junior authors may not fully grasp that using ChatGPT to write a paper without disclosure is an ethical lapse [1, 9]. Clear guidelines akin to plagiarism guidelines should be provided. For example, "Text generated by AI must be treated as third-party content: it should be credited or used only as allowed.", much like one cannot just copy someone else's text without citation. Major conferences and journals could include a statement in their ethics guidelines to this effect.

In summary, our policy recommendations center on transparency, rigorous filtering, and realigning incentives. We believe these measures, if adopted, would greatly mitigate the negative impact of low-value AI-generated surveys. It would ensure that surveys remain a respected form of scholarship, authored by experts who leverage tools but also inject expertise. This balanced approach allows us to stop the DDoS attack without dismissing the potential utility of AI in reducing writers' workload. Crucially, it reinforces a norm that more is not better when it comes to information. Quality over quantity must be the primary principle if we are to navigate the age of AI in scientific publishing without losing our way.

# 6 Dynamic Live Surveys: Embracing the AI Evolution

Apart from suggestions in Section 5, we would like to propose an experimental idea, which is more forward-thinking and revolutionary. We first point out the challenges facing traditional static surveys, and then propose our scheme of *"Dynamic Live Surveys"*.

**Problems of traditional static surveys.** Static, one-off survey articles struggle to keep pace with the rapid evolution of research fields such as NLP, computer vision, and multimodal learning. Year after year, communities are flooded with repetitive or superficial overviews that quickly lose relevance, leaving practitioners and newcomers alike struggling to distinguish signal from noise. The traditional publication cycle (i.e., draft, submit, review, and publish) can span several months, by which time critical breakthroughs may already have shifted the landscape. Moreover, the increasing volume of static surveys adds to cognitive overload, as readers must sift through numerous overlapping documents to find substantive insights.

**Collaborative and Dynamic Live surveys.** To address these challenges and reduce redundancy, we propose to replace the static model with a "Collaborative and Dynamic Live Surveys" framework. It is an open, online knowledge repository that grows and refines itself through seamless integration of AI-driven content ingestion and domain-expert curation.

Under this framework, a community member first establishes a survey topic wiki by specifying the scope, key research questions, and seminal references, which thereby sets a clear thematic boundary and initial structure. Thereafter, an LLM-based ingestion agent continuously monitors preprint archives, conference proceedings, and benchmark leaderboards. It automatically extracts abstracts, figures, and key performance metrics; synthesizes concise summaries of new results; updates the citation graph to reflect inter-paper relationships; and flags emerging research trends for further review. By design, these automated updates occur within hours of publication, ensuring that the repository remains at the cutting edge.

Human contributors then step in to provide the interpretive depth that machines alone cannot offer. They refine evolving taxonomies to capture subtle methodological distinctions, coordinate conflicting interpretations of algorithmic innovations across different subfields, and provide deeper critical comparisons to the document. For example, experts may contrast the sample efficiency of attention-based architectures against retrieval-augmented models, or evaluate the robustness of emerging evaluation metrics under real-world noise. Each contribution, whether an AI-generated draft or a human-authored critique, is recorded with contributor metadata, timestamps, and change diffs. This version control approach ensures full transparency and accountability, while enabling branching of experimental taxonomies that can be merged back into the mainline once community consensus is reached.

Key features of this model include:

- **Real-time updates**: Automated agents scan multiple sources daily (e.g., arXiv, conference proceedings, benchmark leaderboards), so that new algorithms and datasets appear within hours of release, addressing the lag inherent in static publishing.
- **Human–AI curation loop**: Domain experts guide the agent's focus through prompt refinement, validate or restructure taxonomy nodes, and adjust conflicting interpretations, while the agent handles routine ingestion, formatting, and initial summarization.
- **Versioning and branching**: Inspired by software development, branches allow contributors to explore alternative taxonomies, methodological debates, or experimental structures, which are merged into the mainline only after rigorous review and voting.
- **Incentive alignment**: Contributor recognition (via ORCID linkage, digital badges, co-authorship on archived snapshots, or formal citations) rewards both human authors and agent-maintainers for high-value edits, driving continual engagement and community ownership.

By shifting from static articles to a live, evolving ecosystem, Dynamic Live Surveys largely reduce the peer-review and cognitive overload associated with paper spam. Researchers can subscribe to specific subtopics or methodological threads, receiving targeted notifications as relevant entries update, thus streamlining literature discovery and avoiding redundant reading. An intuitive web interface offers linear narrative views, hierarchical outlines, and interactive citation graphs, facilitating both deep

dives into methodological details and rapid navigation across topics. Periodic archived snapshots (for instance, quarterly or semi-annually) generate citable records that preserve the document's evolution for formal citation, tenure review, and historical analysis.

By harnessing AI for routine ingestion and experts for nuanced synthesis, this live ecosystem delivers continuously updated, high-quality surveys tailored to community needs. In doing so, it transforms how knowledge is curated, shared, and advanced—providing an adaptive, scalable, and collaborative model that we believe represents the optimal path forward for scholarly synthesis in the LLM era.

# 7 Alternative Views

## 7.1 AI-Generated Surveys are Valuable

Survey papers have long been important entry points for researchers, offering clear overviews and key references in a field. But with the rise of LLMs, this role is changing. Many researchers now turn to tools like ChatGPT to quickly get a general summary. If a survey paper offers only that same basic summary, its value may seem limited. We believe surveys still matter, but in the age of AI, the standard must be higher. Good surveys should go beyond just listing work. They should explain why certain ideas worked, how different methods relate, and what deeper patterns or insights exist. These are the kinds of understanding that LLMs alone cannot fully provide.

## 7.2 AI-Generated Surveys are New Tools to Augment Research

Throughout history, new technologies have always helped scholars without replacing human expertise. Just like calculators speed up math and search engines help us find information quickly, AI can be leveraged as a powerful research tool. This continues a long tradition of using machines to handle heavy data work, allowing humans to focus on interpreting, summarizing, and thinking creatively.

However, this comparison to past tools has its limits. Modern AI is powerful, but not perfect. It can inherit biases from its training data and may produce misleading results if not carefully checked. AI can also make up or distort information. Moreover, AI doesn't truly understand context or cause and effect like humans do, so it can miss the reasons behind findings. In other words, AI can combine existing knowledge but doesn't create new ideas on its own. Therefore, even if AI speeds up literature searches, researchers must apply critical judgment. Without careful oversight, this "tool" can just as easily mislead us as save time.

# 8 Conclusion

The rise of AI-generated survey papers marks both a technological milestone and a cultural inflection point in scientific publishing. While large language models offer unprecedented capabilities in information synthesis, their unregulated use has catalyzed a troubling new phenomenon: the *survey paper DDoS attack*. The academic community now faces a deluge of low-effort, high-frequency survey uploads that threaten to drown out meaningful contributions, degrade review quality, and compromise scholarly trust. In this position paper, we have argued that this trend cannot be passively tolerated. Instead, we must actively reshape the norms of survey production by imposing rigorous human oversight, mandating transparent disclosure of AI involvement, and transitioning from static, one-off survey articles to collaboratively maintained, *Dynamic Living Surveys*. These actions are not anti-AI; they are pro-integrity. The goal is not to suppress automation, but to integrate it responsibly, enhancing human insight rather than replacing it. Ultimately, if surveys are to remain trusted instruments of knowledge curation, we must reject the notion that quantity can substitute for quality. In an era where content can be generated at scale, the true challenge lies in sustaining meaning, depth, and community accountability. We believe that with thoughtful design and principled intervention, the future of AI-assisted surveying can be one of collaboration, not collapse.

# Acknowledgments

This paper is supported by National Natural Science Foundation of China (624B2096, 62322603).

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

# A  More Evidence

## A.1  Observation From Empirical Metrics

**Citation Overlap.** We investigate the citation overlap of the CS survey papers. Specifically, we sample several papers that are related to similar topics for each year, respectively, and compare their citations. The observations are summarized as follows:

- Citation overlap percentage is $< 40\%$ for pre-2022 and $> 60\%$ for post-2022, with $< 1\%$ for a random baseline between two arbitrary surveys.
- Jaccard Index for citation overlap is $< 0.3$ for pre-2022 and $> 0.5$ for post-2022.

**Similarity Scores.** We use acge-text-embedding[3] as the text embedding model, and encode the abstracts of surveys with similar topics. The score increased significantly from 0.6033 in 2022 to 0.8367 in 2023, and subsequently stabilized at a comparable level of 0.7986 in 2024. We can observe that the semantic similarity of the selected surveys views a clear "post-2022 spike", showing the semantic repetition of the survey content.

## A.2  Observations From Other AI Detectors

To cross-validate the observed post-2022 spike in AI-generated survey papers, we use two more recent detectors from top-tier conferences: DeTeCtive [10] and MAGE [16]. While absolute scores vary across different detectors, the crucial trend identified in our paper remains remarkably consistent. Specifically, incremental ratios of the AI-generated scores from 2020 to 2025 are shown in Table 1.

Table 1: The growth ratio of AI-generated CS survey papers from 2020 to 2024 detected by DeTeCtive and MAGE.

| Method | 2020-2021 | 2021-2022 | 2022-2023 | 2023-2024 |
|---|---|---|---|---|
| DeTeCtive | 0.2337 | 0.1060 | 0.3081 | 0.4210 |
| MAGE | 0.1586 | 0.1860 | 0.7058 | 0.5300 |

Combined with the original method, all three detectors show a significant acceleration in AI-generated scores post-2022, strongly corroborating our central finding of a "post-2022 spike". This consistent trend provides robust evidence for our position.

# B  Link between DDoS Attacks and Survey Paper Flooding

We use the "DDoS attack" analogy to frame the issue as a systemic problem, not just "too many papers". The flood of low-quality content overwhelms the community's finite attention and peer-review capacity. This effectively denies service to researchers seeking genuine scholarly insight, forcing them to wade through "literature clutter".

As detailed in Section 4, this also burdens the peer-review system and risks "literature poisoning". The framing argues that the sheer quantity has become a substantive barrier to research, not a minor inconvenience. We believe this is a critical position for the community to consider.

---

[3]https://huggingface.co/aspire/acge_text_embedding

