# OpenReview forum: "Stop DDoS Attacking the Research Community with AI-Generated Survey Papers"
_NeurIPS.cc/2025/Position_Paper_Track — NeurIPS 2025 Position Paper Track Oral_

### Official Review · Reviewer_wDy1 · 2025-07-04

**Significance:** 4
**Presentation:** 4
**Rating:** 7
**Confidence:** 5

**Summary:**

This paper argues that the unchecked growth of AI-generated survey papers, often redundant, superficial, or hallucinated, is flooding preprint platforms and harming the quality, trust, and usability of academic literature. The authors term this trend a "survey paper DDoS attack." Through empirical evidence from arXiv (e.g., rising paper volume, increased AI-content scores, and abnormal submission patterns), the paper shows a post-ChatGPT surge in questionable survey output. It raises concerns about quality, ethics, and scholarly value. As a solution, the authors call for transparency in AI usage, stronger peer review norms, and propose a new model: "Dynamic Live Surveys," collaboratively maintained and version-controlled repositories integrating AI tools with expert oversight.

**Strengths:**

The paper takes a strong and timely stance on a rising issue with well-substantiated arguments. It combines empirical data with cultural analysis, presents a novel metaphor (“DDoS”), and proposes actionable recommendations and a forward-looking alternative (Dynamic Live Surveys). Its tone is measured, critical of misuse, but not anti-AI.

**Weaknesses:**

While well-argued, the paper could further explore how to reliably distinguish between responsible and irresponsible use of LLMs, especially given the detection limitations it mentions. The “Dynamic Live Surveys” proposal, though compelling, remains largely conceptual and may benefit from a more concrete feasibility discussion. Furthermore, it could engage more with how different disciplines outside CS have handled similar synthesis challenges.

**Questions:**

- How might Dynamic Live Surveys be governed in a decentralized yet quality-assured way? How do we real-time benchmark it in Production environment ?
- Is there a risk of overcorrecting and discouraging useful AI-assisted contributions? How can that balance be maintained?
- Could the proposed detection heuristics be integrated into arXiv moderation or conference review pipelines practically and automated ?

**Alternative Position:**

Yes, and alternative positions are well-considered and addressed by the argument

**Author Identification:**

No.

**Context:**

3

**Discussion:**

3

**Ethics:**

["NO or VERY MINOR ethics concerns only"]

**Position:**

Yes, the paper argues for or against a position related to machine learning.

**Support:**

4

**Thoroughness:**

5

---

### Official Review · Reviewer_Nge9 · 2025-07-06

**Significance:** 4
**Presentation:** 3
**Rating:** 4
**Confidence:** 4

**Summary:**

This paper raises an important issue of AI-generated surveys as a DDOS attack on academia.

**Strengths:**

First off I think this is an interesting paper.
What I like
1. The idea of the living survey
2. The definition of abnormal authors
3. The potential detection metrics (section 3.2)

**Weaknesses:**

to what extent does the AI-generation score actually predict AI-generated content? The concern raised from two issues:

1. There have been some papers talking about LLM writing detection tool not being reliable https://direct.mit.edu/qss/article/doi/10.1162/qss_a_00368/128867/Where-there-s-a-will-there-s-a-way-ChatGPT-is-used and thus researchers have developed their own tools based on LLM-revising texts (ground truth 1), see https://www.cell.com/patterns/fulltext/s2666-3899(23)00130-7 and https://arxiv.org/abs/2404.01268 and a lot of others. I looked into the tool authors used and I found that https://huggingface.co/desklib the lab that developed this tool is not that reliable.

2. ChatGPT was released at the end of 2022 which arguably started all these "ddos attacks". If the argument holds true, then we should observe a huge increase in AI-generated surveys across multiple measures, a strong increase compared to 2022. OK i see this from the figure 1 mid panel which is nice, but for the left and right panels I don't see this trend which makes me further concern the reliability of the conclusion and reliablity of the measures (abmormal authors etc) -- they are nice measures and make sense intuitivly but not experimentally.

**Questions:**

How exactly are DDoS attacks linked to survey papers? it is less clear -- my feeling is just that we have too many AI generated survey papers, ok i buy it -- but authors claimed that "Instead of helping the research community to digest a field, it starts to **feel like** a DDoS attack, flooding the field with more content than anyone can reasonably read or use." I don't think it is a substantive link, and framing this to a DDOS attack seems to be overselling

**Alternative Position:**

Yes, and alternative positions are trivial straw-man arguments

**Author Identification:**

No.

**Context:**

3

**Details Of Ethics Concerns:**

no ethics concerns

**Discussion:**

3

**Ethics:**

["NO or VERY MINOR ethics concerns only"]

**Position:**

Yes, the paper argues for or against a position related to machine learning.

**Support:**

3

**Thoroughness:**

4

---

### Official Review · Reviewer_sdPS · 2025-07-31

**Significance:** 4
**Presentation:** 3
**Rating:** 8
**Confidence:** 4

**Summary:**

This position paper argues that the research community is currently being flooded by AI-generated survey papers, since LLMs such as ChatGPT came to mainstream use. The authors refer to this as "DDoS attack on survey papers", comparing to distributed denial-of-service attacks, when a large volume of mediocre quality inputs crowds out meaningful work. Through empirical analysis of arXiv data specific to CS community including citation patterns and author behaviors the paper show that there is a post-2022 spike in AI-generated surveys, many of which lacks quality. The paper also critiques the motivations behind such papers and discusses their impact on trust we have as a community on research, associated stress on reviewers and overall culture. It proposes some solutions including stricter reviewing standards, AI-detection protocols and also the need for Dynamic Live Surveys as an alternative to current standard reviewing methods.

**Strengths:**

The issue of spike in AI-generated review papers has been lunch-table conversation in academia in the past few years. This paper is timely and raises systematically a pressing issue we have in the research community and puts in the context of a position paper. The topic is therefore widely relatable and of course somewhat controversial. The authors argue their opinion clearly with empirical evidence. The paper also does well to anticipate counterarguments and handle them reasonably.

**Weaknesses:**

The analysis of the paper is based on simple heuristics such as counting papers with certain keywords, checking for overlapping citations and using AI-content detectors. While this may be reasonable for a position paper, the paper claims may be strengthened using better methods:

- Compared the citation overlap against a random baseline to see what counts as abnormally high
- Use similarity scores or similar to measure how repetitive the content actually is
- Also what about false positives such as single authors who wrote multiple high-quality surveys since they are experts in the field, this is a possibility as well.

Additionally, the paper does not appear to address alternative methods to approach the problem:
- Should we be encouraging multimodal, code-linked or interactive alternatives instead?
- For example, are "reviewed" presentations a better format overall long form text-heavy survey papers?
- Should there be review papers at all?

**Questions:**

The proposal for Dynamic Live Surveys is interesting, but does not go into the deeper implementation challenges:
- For example, how will hallucinated content from LLMs be caught here?
- How will incentives and moderation be handled at scale?
- Who oversees the taxonomic revisions?  What happens when there conflicting edits?
- How can hallucinations or biased outputs be controlled for dynamic live surveys?
- Are human reviewers re-verifying every update?

**Alternative Position:**

Yes, and alternative positions are well-considered and addressed by the argument

**Author Identification:**

No.

**Context:**

3

**Discussion:**

4

**Ethics:**

["NO or VERY MINOR ethics concerns only"]

**Position:**

Yes, the paper argues for or against a position related to machine learning.

**Support:**

3

**Thoroughness:**

4

---

### Note · Authors · 2025-09-05

**1-11 Submit Again:**

Definitely yes

**1-1 Submission Process:**

4

**1-2 Next Year:**

For the next year, we would like to suggest the following things:
- Introduce a focused author-response phase for clarifying claims and scope to efficiently resolve misunderstandings.
- Feature a live panel or debate at the main conference with authors of the most impactful papers to discuss the problem together.
- Establish a public topic crowdsourcing platform to gather points of interest from all sectors before the submission period, thereby guiding papers to focus on the most cutting-edge and influential directions.

**1-3 Future Development:**

We are willing to offer some suggestions to help improve the NeurIPS Position Track:
- Clarify Scope: publicly define it as a venue for provocative arguments and future roadmaps, not surveys or incremental research.
- Provide a Calibration Kit: give reviewers explicit criteria (e.g. vision, argument strength, impact), and exemplar reviews and links to past accepted papers.

**1-4 Interest:**

["Panel discussions with other position paper authors", "Structured debates on controversial topics", "Workshops for developing position papers", "Mentorship programs for early-career researchers"]

**1-5 Thoughtful:**

9

**1-6 Supportive:**

10

**1-7 Technical Aspects Versus Position:**

8

**1-8 Gate Keeping:**

10

**1-9 Camera Ready Changes:**

If our paper is accepted, we will make the following key revisions to significantly strengthen the manuscript for the camera-ready version:

1. Strengthen the Empirical Evidence:
- Cross-Validation of AI Content Detection: We use two additional SOTA detection tools (DeTeCtive, NeurIPS 2024; MAGE, ACL 2024), and validate that "post-2022 spike" is a consistent and robust trend observable across multiple detectors, solidifying our core position.
- More Year-on-Year Evidence: We display annual growth rates for counts of survey papers and "abnormal authors", further validating and visualizing the post-2022 acceleration trend.

2. Elaborate on "Dynamic Live Survey" Proposal:
We will expand the discussion on its feasibility and governance. We will detail a concrete framework for the Dynamic Live Survey built upon three pillars: (1) a Human-AI Curation Loop for initial quality control, (2) a Git-like Version Control System for decentralized peer review and quality assurance, and (3) mechanisms for Real-time Updates from sources like arXiv. This will transform the proposal from a high-level concept into a more tangible and actionable vision.

3. Refine Key Arguments and Scope:
- We will sharpen the "DDoS attack" analogy to more explicitly link the flood of content to a "denial of service" on the finite attention of the research community.
- We will strengthen our discussion on maintaining a healthy balance in AI usage, emphasizing mandatory disclosure as a practical policy for fostering transparency and accountability.
- We will clarify that while our analysis is focused on Computer Science as the epicenter, the principles and framework we propose offer a valuable starting point for other disciplines facing similar challenges.

These revisions will not change our paper's core position but will significantly enhance its empirical foundation, the feasibility of its proposals, and the overall clarity of its message.

**3-1 Review Response1:**

sdPS

**3-2 Reaction To Review1:**

The reviewer provides thoughtful comments and strong support for our position, raising several insightful points that will further improve our paper. We feel fully inclusive, not gatekept. We summarize the review and give our responses:

>W1.1 & 1.2 (Citation overlap & Similarity)
We conduct further analysis to validate our "post-2022 spike" on AI-generated surveys. More details will be included in our revised version.
- Citation overlap percentage is <40% for pre-2022 and >60% for post-2022, with <1% for a random baseline between two arbitrary surveys.
- Jaccard Index for citation overlap is <0.3 for pre-2022 and >0.5 for post-2022.
- Semantic similarity scores also witness a spike from ~0.70 for pre-2022 to ~0.84 for post-2022, evaluated by acge-text-embedding for abstracts from surveys with similar topics.

>W1.3 (False Positives for Abnormal Author): We agree that this is a valid concern. Our "abnormal author" metric uses a deliberately high threshold (>3 surveys in 1 month with less than 2 co-authors) that makes false positives for genuine experts highly unlikely, as this pace is unsustainable for high-quality, critical work. The spike of abnormal authors also validates our position.

>W2 (Alternatives): We believe high-quality surveys are more vital and indispensable than ever, precisely because of the flood of low-quality, redundant AI-generated content. Our "Dynamic Live Survey" is proposed as an evolution of the format—an interactive, multimodal, community-maintained resource designed to preserve the genre's integrity. We will clarify this vision.

Qs (Live Survey): The core of our proposal is a Human-AI curation loop to validate content and mitigate hallucinations. Quality control and moderation will be managed via a version-control system (e.g., branching for major edits, community review before merging). We propose incentives such as ORCID linkage, digital badges, and co-authorship on citable snapshots to encourage expert contributions.

**3-3 Review Response2:**

Nge9

**3-4 Reaction To Review2:**

We appreciate that the reviewer agrees with our position and finds it valuable. The proposed weakness primarily focuses on more experimental support, which might extend beyond the typical scope of a position paper. But we are happy to give more evidence, which further strengthens our position.

>W1: Reliability of AI-generation scores
To cross-validate our findings, we use two recent detectors from top-tier conferences: DeTeCtive (NeurIPS 2024) and MAGE (ACL 2024). Combined with our original method, all three detectors show a significant acceleration in AI-generated scores post-2022, strongly corroborating our central finding of a "post-2022 spike." This consistent trend provides robust evidence for our position.

>W2: Clarity of trends in other metrics
To make the trends more explicit, we analyzed the annual growth rates. The results reveal a clear acceleration point after 2022, strongly supporting our hypothesis.
- CS Survey Papers: The growth rate nearly doubled from 9.8% (21-22) to 17.4% (22-23).
- Abnormal Authors: The growth rate nearly tripled from 8.4% (21-22) to 24.0% (22-23).
These findings are also consistent with independent studies reporting exponential growth in AI publications (e.g., arXiv:2508.04586). We will revise our figures to display these growth rates, making the trend undeniable.

>Q: Link between DDoS attacks and survey papers
We use the "DDoS attack" analogy to frame the issue as a systemic problem, not just "too many papers." The flood of low-quality content overwhelms the community's finite attention and peer-review capacity. This effectively denies service to researchers seeking genuine scholarly insight, forcing them to wade through "literature clutter."

As detailed in our paper, this also burdens the peer-review system and risks "literature poisoning." The framing argues that the sheer quantity has become a substantive barrier to research, not a minor inconvenience. We believe this is a critical position for the community to consider.

**3-5 Review Response3:**

wDy1

**3-6 Reaction To Review3:**

We are grateful for this thoughtful and supportive review, which recognizes the novelty, clarity, and timely stance of our position. We feel this is exactly the kind of constructive, non-gatekeeping dialogue this track aims to foster. We appreciate the opportunity to elaborate on these excellent points.

> Practical Integration & Distinguishing AI Use (W1&Q3)
We agree that reliably distinguishing responsible AI use is a core challenge. This is precisely why our paper proposes behavioral and statistical heuristics (Sec 3.2) as a practical first step, rather than relying solely on fallible content detectors. These heuristics are designed for practical integration, and can act as robust flags for human review, helping to manage the overwhelming scale of submissions, as discussed in recent reports [1].

> Dynamic Live Survey (W2&Q1)
The reviewer raises an excellent point about feasibility. We offer the Dynamic Live Survey as a forward-looking vision, with governance built on three key pillars:
- Human-AI Curation Loop: Domain experts guide and validate AI-ingested content, ensuring quality from the start.
- Version Control for Quality: A Git-like system with branching and community peer-review for major edits would ensure decentralized yet rigorous quality assurance before merging changes.
- Real-time Updates: Automated agents would scan sources like arXiv and leaderboards daily, allowing updates to appear within hours, not months.

> Healthy Balance for AI usage (Q2&W3)
Our goal is not to discourage useful AI assistance but to foster responsible and transparent use. The key is a policy of mandatory disclosure. Like existing "LLM Usage" policies, authors should declare how AI tools were used. This maintains human accountability while leveraging AI as a tool. While our paper focuses on CS as the epicenter of this issue, we agree this principle of transparency offers a valuable starting point for a broader conversation across many disciplines.

---

### Meta-Review · Area_Chair_EGqF · 2025-08-31

**Rating:** 7
**Confidence:** 4

**Strengths:**

The reviewers all like the idea of a living survey. They found the definition of an abnormal author thought provoking. The appreciated the detection metrics. They all agree that the paper is timely and raises systematically a pressing issue we have in the research community. The topic is therefore widely relatable and of course somewhat controversial. The authors argue their opinion clearly with empirical evidence and take a strong stance. The paper also does well to anticipate counterarguments and handle them reasonably.

**Weaknesses:**

The reviewers pointed out that the simple heuristics (counting papers with certain keywords, checking for overlapping citations and using AI-content detectors) is not the best empirical evidence and suggest improvements.
They also point out that alternative methods to approach the problem, which I see a discussion already happening. The Dynamic Live Survey though compelling, remains largely conceptual and may benefit from a more concrete feasibility discussion.

**Questions:**

It would help if the alternative approaches consider the following questions

* How will hallucinated content from LLMs be caught here?
* Does the paper need the DDoS reference?
* What is the governance approach to Dynamic Live Surveys?
* How could disclosure requirements avoid discouraging useful AI-assisted contributions?
* Could the proposed detection heuristics be integrated into arXiv moderation or conference review pipelines practically and automated?

**Ethics:**

No ethics concers were raised by any of the reviewers.

**Thoroughness:**

3

---

### Decision · Program_Chairs · 2025-09-26

Accept (Oral)